# Phages and Human Health: More Than Idle Hitchhikers

**DOI:** 10.3390/v11070587

**Published:** 2019-06-27

**Authors:** Dylan Lawrence, Megan T. Baldridge, Scott A. Handley

**Affiliations:** 1Division of Infectious Diseases, Department of Medicine, Washington University School of Medicine, St. Louis, MO 63110, USA; 2Edison Family Center for Genome Sciences & Systems Biology, Washington University School of Medicine, St. Louis, MO 63110, USA; 3Department of Pathology and Immunology, Washington University School of Medicine, St. Louis, MO 63110, USA

**Keywords:** metagenomics, microbiome, bacteriophage, virome, genomics

## Abstract

Bacteriophages, or phages, are viruses that infect bacteria and archaea. Phages have diverse morphologies and can be coded in DNA or RNA and as single or double strands with a large range of genome sizes. With the increasing use of metagenomic sequencing approaches to analyze complex samples, many studies generate massive amounts of “viral dark matter”, or sequences of viral origin unable to be classified either functionally or taxonomically. Metagenomic analysis of phages is still in its infancy, and uncovering novel phages continues to be a challenge. Work over the past two decades has begun to uncover key roles for phages in different environments, including the human gut. Recent studies in humans have identified expanded phage populations in both healthy infants and in inflammatory bowel disease patients, suggesting distinct phage activity during development and in specific disease states. In this review, we examine our current knowledge of phage biology and discuss recent efforts to improve the analysis and discovery of novel phages. We explore the roles phages may play in human health and disease and discuss the future of phage research.

## 1. Introduction

Bacteriophages were independently discovered more than one hundred years ago by both Frederick Twort and Félix d’Herelle [1,2]. Since their discovery, phages have played a key role in our understanding of biological processes and have served as essential tools in many seminal biological works. However, more than a century later, we have only recently begun to appreciate the diversity and complexity of phages. Most known phages, and specifically 96% of phages examined by electron microscopy, belong to a single order, *Caudovirales*, or tailed phages [3]. Until recently, our knowledge of phage diversity has been limited to phages isolated by plaque assay, a method of culturing phages [4]. This method is robust but requires both a culturable bacterial host and a phage with a lytic lifecycle. Modern nucleic acid sequencing technology has allowed for the discovery of novel phage genetic material, but the majority of these putative phage sequences cannot be classified. Termed “viral dark matter,” more than 80% of virally derived DNA is still a mystery [5]. Coupled to this is the challenge of identifying bacterial hosts for phages [6]. While often considered exquisitely specific, studies have increasingly found promiscuous phages infecting bacteria across orders [7]. These challenges provide numerous opportunities for novel research approaches, but should also serve as a reminder of the daunting task ahead of virome researchers.

In the past two decades, there has been an explosion of research and interest in the human microbiota, particularly in the gut [8]. This research has primarily focused on the bacterial portion of the microbiota, but more recent work has begun to examine the human virome, wherein phages comprise the majority of classified sequences [9]. Studies using human stool samples, discussed in detail below, have suggested important roles for endogenous phages in microbiota development and dysbiosis. Additionally, recent studies have shown phages are not merely players in the human gut but are likely important at other sites of the human body [10].

Since the discovery of phages, there has been considerable interest in their use as therapeutic agents due to their ability to modify or destroy bacteria. Termed “phage therapy”, the use of phages to treat drug-resistant bacterial infections has stimulated interest in the use of phages to treat a variety of human diseases [11,12]. However, given our incomplete understanding of the complexity of phage biology, questions remain as to whether phage therapy would be a safe treatment modality. Often overlooked in reviews of phage therapy are the important interactions of phages and bacteria in the healthy human gut, dynamics which could be altered by the introduction of foreign phages.

In this review, we present the current knowledge of phage biology and genetics, focusing on those genetic elements unique to phages. We then discuss current tools for detection and identification of novel phages in mixed samples and highlight some of the drawbacks and limitations of current methods. Then, we turn our attention to human health, discussing what is currently known about phages and their role in human development and disease. Finally, we look towards the future of phages both from a research and therapeutic perspective and discuss roles phages could play in the future of medicine and what discoveries are on the horizon.

## 2. Phage Biology

It is estimated that there are nearly 10^31^ viruses on Earth, the vast majority of which are phages [13]. Here we present a brief overview of phage biology; several reviews and other sources have covered this topic in greater depth [14,15]. Traditionally, phages have been grouped into either lytic or lysogenic life cycles (Figure 1). Lytic phages infect their target bacterial host and very quickly begin replication of viral progeny. When sufficient numbers of progeny are produced, the host cells lyse, generally killing the host in the process. In contrast, lysogenic phages undergo very little replication in their host. During infection, lysogenic phages instead incorporate their genome into that of their bacterial host, or may maintain it as a plasmid [16] where it remains and is passively replicated when the bacterium replicates. These phages are referred to as “prophages.” However, lysogenic phages are able to become lytic with exposure to certain environmental stimuli [17]. For example, antibiotics and host inflammation, two common features of modern human life, can both induce lytic lifecycles in phage [18,19]. 

More recently, two additional phage lifecycles have been observed (Figure 1). Pseudolysogenic phages have been described as being in a state of “stalled development”, and can either be incorporated into the host genome or remain as free DNA in the cytoplasm [17]. This free DNA is distinct from the plasmid form of some lysogenic phages [20]. Their genetic material is inserted into the host cell, but rather than incorporating into the host genome, the genetic material remains free in the cytosol. These phages can undergo lytic or lysogenic lifestyles depending on environmental conditions. Pseudolysogenic phages may also be referred to as being in a phage “carrier state” [21,22]. The final type of lifecycle is referred to as chronic infection, which is similar to a lytic lifecycle as the phage is actively replicating in the host and producing viral progeny. However, unlike in lytic lifecycles, chronic phages do not lyse their host, but rather, the viral progeny is exported by a variety of mechanisms [20]. 

Phages are classified according to the International Committee on Taxonomy of Viruses (ICTV) largely based on genetic and protein sequence homologies [23]. The most well-classified order of phages, because these were among the first to be carefully studied, is *Caudovirales* which is further divided into five families, the three most well-characterized being *Myoviridae, Siphoviridae,* and *Podoviridae*. There are also two more recently described families: *Ackermannviridae* and *Herelleviridae* [24,25]. Separated initially by morphological differences, *Caudovirales* are all double-stranded DNA (dsDNA) phages marked by the presence of a tail. Over time other types of phages, including the filamentous single-stranded DNA (ssDNA) phages *Inoviridae* and the ssDNA *Microviridae* families were also described. ICTV currently defines 19 phage families, but the ones highlighted above represent the most well-characterized.

The classification of phages is far from straightforward. Though some genes have been defined as “phage hallmark genes”, these features are not present in all phages [26]. In fact, some of the “hallmarks”, such as terminases, portal proteins, tail proteins, and major capsid proteins, are derived almost exclusively from studies of *Caudovirales*. In general, studies focus on identifiable phages, which are almost always primarily *Caudovirales* or *Microviridae,* with few studies targeting other phages. However, some recent work has been done on the discovery of RNA phages by performing analyses on previously generated metagenomics datasets; these studies have identified a trove of novel RNA phages [27]. Thus, many additional phages may remain to be discovered.

## 3. Isolation and Genomic Analysis of Phages

### 3.1. Isolation of Phages

Traditionally, phages have been isolated using plaque assays, performed by growing a lawn of bacteria on an agar plate and then introducing a suspension suspected of containing phages [4] (Figure 2). Often phages in suspension must be enriched beforehand, which can be done by inoculating a liquid culture of bacteria with the suspected phage and allowing the culture to clarify [28]. The plate is allowed to incubate and is then examined for plaques (clearings in the bacterial lawn), which represent locations in which the bacteria have been lysed. Once plaque-isolated, sequencing of a phage genome is relatively trivial, given our ability to bioinformatically assemble single organisms. The advent of long-read sequencing using systems such as PacBio or Oxford Nanopore has allowed for highly accurate *de novo* assembly of bacterial isolate genomes, which was previously challenging with only short-read Illumina-based sequencing [29]. Long-read technology has recently been applied to sequencing phage isolates with great success [30], as well as to environmental samples where long-read approaches provide improved assemblies with similar abundance information to short-read sequencing [31]. The concerted efforts of classical phage isolation via plaque assay in combination with modern sequencing technologies will undoubtedly fill many holes on our genomic knowledge of bacteriophage. However, plaque assays have several caveats making them inefficient for comprehensive phage discovery efforts. One of the biggest drawbacks to plaque assays is their requirement for a culturable host, as many species of bacteria are currently challenging to cultivate. Additionally, plaque assays are only able to capture phages which form plaques, which may require additional treatments, such as mitomycin C, to induce phage into a lytic state [32]. These caveats help explain why we have only discovered phages infecting 3 of the 61 known bacterial phyla [33].

Given the challenges of plaque assays, several alternative methods have been employed for phage discovery. An increasingly widespread approach is metagenomic sequencing [35]. Prior to metagenomic sequencing, genetic material is enriched and collected via virus-like particle (VLP) preparation. VLP preparation is a broad descriptor which generally involves removal of bacteria and free nucleic acid from a sample, thereby enriching for viral particles, which are then broken open to isolate DNA or RNA. Most VLP methods focus on extraction of DNA. Today there are a number of chemical and physical isolation methods for VLP preparations including cesium chloride gradients, microfiltration, and centrifugation [36]. Recently iron chloride flocculation has also been used to enrich for phages in complex samples [37]. After phage genetic material has been isolated, it is typically Illumina sequenced to generate large numbers of short, individual sequences. Sequence data is then assembled into contigs or complete genomes. A similar approach can be used for RNA phages by performing a cDNA synthesis step before sequencing. Metagenomic sequencing is not without its downsides, and chief among these is the aforementioned challenge of dark matter, as it can be challenging to assemble genomes and/or assign taxonomy to individual reads in the absence of reference genomes, though these issues may be addressed by improvements to databases and search and assembly algorithms. In addition, metagenomic assembly is imperfect and may produce chimeras, misleading interpretation. Thus, while metagenomic sequencing enables an unprecedented approach to exploring phage diversity, results should always be interpreted with consideration of these challenges.

More recently, single amplified genomes (SAGs) and virus-SAGs have been used for phage discovery [38,39]. SAGs represent a potentially powerful tool for both the discovery of phages and the ability to derive specific bacterial hosts for phages. To perform this method, fluorescence activated cell sorting (FACS) is applied to a sample of interest to sort and analyze individual bacteria. This method has been used to capture full bacterial genomes as well as to uncover phage virions in the cytoplasm or adhered to the surface of these cells and thereby isolate phage genomes directly [38,40]. Virus-SAGs are an alternative approach to the method above in which viral particles are sorted directly, thus eliminating most host information [41]. As with any method of analyzing complex communities, current SAG techniques are biased toward recovery of more common community members. While still relatively novel in terms of phage research, these methods may open the door to in-depth studies of bacteria–phage interactions that were previously not possible. Finally, chromatin conformation capture (3C) is another recent approach for metagenomic analysis that can be applied to phage discovery. 3C is a method of examining long-range DNA interactions and a modification to this, meta3C, has been used to derive phage–bacteria interactions in the human gut [42]. The opportunities to clearly define relationships between phage and their bacterial hosts, as well as to obtain novel phage genomes, are thus continuing to develop.

### 3.2. Computational Identification of Phages

Perhaps even more challenging than isolation of phages is the genomic analysis that follows. *De novo* identification of phages is a challenging computational problem. The past two decades have had several pieces of software published presenting novel approaches to this problem. Phage_Finder was one of the first programs designed for phage discovery [43]. This was followed by Prophinder, PhiSpy, PHAST, and PHASTER., which all worked to build upon and improve discovery of prophages in bacterial sequencing data [44,45,46,47]. However, none of these programs were designed to discover non-integrated phage sequences in large-scale data sets. In the past five years, there has been concerted effort to create tools for the discovery of non-integrated phage in metagenomic data. Both VirSorter and VirFinder were designed specifically for the identification of non-integrated phage sequences [5,33]. These programs have been applied to a variety of datasets, including metagenomic data sets, sequences derived from VLP preparations, and SAG datasets. However, even with these advancements, much of what contributes to viral dark matter remains a mystery. 

An examination of the software alone, however, does not fully capture the difficulty in studying phage biology. Early studies on phages in the human gut were done using VLP. These methods isolate both eukaryotic viruses and phages and have revealed the gut virome to contain numerous phages [48]. VLP-based studies have largely focused on DNA phages primarily recovering dsDNA *Caudovirales* as well as a large fraction of ssDNA phages. However, some of these early studies made use of multiple displacement amplification (MDA), which has more recently been shown to bias amplification of ssDNA phages [49,50]. MDA is valuable for analysis of low biomass samples, but in addition to preferentially amplifying small ssDNA viruses, this method is also blind to RNA viruses. Together these studies have generated a staggering amount of viral DNA sequence, but due to poor taxonomic representation in reference databases (of 5639 entries on NCBI Virus, *Caudovirales* represents 4668 entries or more than 80% of sequences), discovery and knowledge of phage genomics is subject to extreme reference bias [51].

Differentiating phage genomes from bacterial genomes can also be challenging due to the co-evolution of phages and their hosts. Phages often have similar GC content and codon usage to their hosts [52]. Given this difficulty, other methods have been derived for the identification of phages in complex samples. Early work on novel phage discovery focused on the identification of phage attachment sites, the locations where phages integrate into the bacterial genome [43]. This is a robust method, but can only identify prophages. A second early approach to identification involved statistical analysis for the enrichment of phage-like genes [44]. However, these approaches are all alignment-based, and suffer heavily from reference database bias, often resulting in only identification of *Caudovirales*-like phages. More recent methods have begun to rely on k-mer analysis where an identity for a phage is defined by its k-mer usage [5], [33]. A k-mer is simply a “k” length stretch of nucleotides, wherein a codon would be a 3-mer. While subtle, these studies revealed unique viral k-mer signatures allowing for identification of phage sequences in mixed samples. K-mers have also been used to identify putative host bacteria for some phages [53]. Other genomic features that can be considered when examining samples for phage sequences are the presence of “hallmark” phage tail genes, limited strand switching of genes, and enrichment of unidentifiable proteins [33]. However, even given these features, identification of phages remains a challenging problem. With increasing appreciation of how important these phage populations can be, the number of research groups focused on resolving these issues continues to increase, which ultimately may yield creative solutions and improved databases for phage classification.

## 4. The Role of Phages in the Human Microbiota

As our knowledge of the bacterial populations that inhabit different niches in the human body has dramatically expanded over the past several decades, researchers have also begun to consider the role phages play in different human microbiota (Figure 3). Here, we discuss what is currently understood about phage in these niches.

### 4.1. Phages in the Infant Gut

Unsurprisingly, the most well-studied area of the human “phageome” is the gut, with most studies focusing on fecal samples. Studies in both healthy humans and those with chronic and inflammatory diseases have uncovered interactions between phages, bacteria, and their human hosts. This complex interplay begins at birth. Intriguingly, at a timepoint when the human gut is relatively sterile with regard to bacterial populations, phage populations in early-life fecal samples are massively expanded in comparison to healthy adults [74]. Longitudinal analysis of infants from time of birth to 24 months of age shows a conserved trend in loss of phage richness over time, as well as a correlation between loss of *Caudovirales* phages and gain of *Microviridae* species, at the same time that bacteria are increasing in richness. This strong negative correlation between bacteria and phages in the infant virome is still not fully understood. Interestingly, while by six months of age, mode of birth could not be derived purely by using analysis of the bacterial microbiome from collected samples [75], even at one year of age, the composition of the gut phageome could be used to determine the mode of birth for infants [62]. These data suggest a key role for vertical transmission of viruses in early microbiota development and in the gradual definition of a mature microbiota composition. 

### 4.2. Phages in the Adult Gut

A majority of studies of the adult phageome have focused on the association of phages with human disease, limiting our knowledge of a “healthy” phageome. Studies of bacteria in the human gut have shown high degrees of inter-personal variation, so much so as to allow fingerprinting of an individual by their gut microbes [63]. However, variation between phageomes of individuals has been shown to be even greater than that of bacteria [64]. Even with this extreme variability, the phageome appears to be highly stable, perhaps more so than the associated bacteria in the gut, even in the presence of strong modifications to the microbiota, such as fecal microbiota transplant (FMT) [65]. Given their ability to evolve rapidly, the human gut likely contains dozens of “novel” phage species within a single individual. However, even with this high degree of mutation and evolution, it appears a “core virome” exists in the gut, providing temporal stability by interacting with the core bacteria found across humans [66]. Phages in the oral microbiota, another important component of the overall gut microbiota, appear to play a role in both homeostasis and disease. The role of phages in the oral microbiota appears to be complex and diverse with both positive and negative relationships found between bacteria and phages [54]. Analysis of broad interactions between bacteria and phages in the human oral cavity revealed a network of cross-infective phages with broad host tropism [55]. These cross-infective phages were found to be positively correlated with commensal bacteria but negatively correlated with bacteria known to cause periodontal disease, indicating a potential protective effect by resident phages in the mouth. This temporal stability of phages in the oral cavity appears to be similar to the stability of phages in the distal gut, which may suggest a broad role for phages in microbiota regulation.

The association of phage with intestinal disease has been explored in several recent studies. Inflammatory bowel disease (IBD) is a broad term including several disorders involving chronic inflammation of the intestinal tract. Two types of IBD are ulcerative colitis (UC) and Crohn’s disease (CD). Both of these conditions are characterized by long-lasting inflammation and periods of increased inflammation and symptoms, referred to as flares. Dozens of studies over the past decade have revealed a broad loss of bacterial diversity in IBD patients [67,76]. Analysis of a longitudinal cohort of IBD patients revealed that in contrast to a decrease in bacterial diversity, there is an expansion of phages in IBD patients as compared to their healthy household controls [77]. Specifically, while both UC and CD patients exhibit increased phage richness, these different IBD subtypes are associated with distinct phage populations. Analysis of correlation between bacteria and phages showed an inverse correlation between specific bacterial taxa and *Caudovirales* phages, a correlation that was stronger in CD than UC patients. Overall, this study indicated potentially important disease-specific roles for bacteria–phage, or possibly phage–host interactions in IBD. Studies in mice have recapitulated these findings. A recent study in mice deficient in adaptive immunity revealed an inverse relationship between bacteria and phages during T-cell induced colitis [78]. Further, human IBD-derived viral contigs recruit more phage reads from colitic mice than from healthy mice, indicating that colitic mice have similar phage alterations as humans with IBD. Together, these studies define a disease-associated relationship between gut bacteria and phage in both humans and mice.

Equally ill-defined are the interactions of phages and eukaryotic cells in the human gut. Treating germ-free mice with a mixture of phages or phage DNA stimulates IFN-γ [79]. This response is mediated through TLR-9, a nucleotide-sensing receptor involved in several immune response pathways, and leads to exacerbated colitis. Additionally, among IBD patients receiving FMT, patients with strong therapeutic responses were found to contain less *Caudovirales* DNA than non-responders. It has also been shown that phages are continuously transcytosed by gut epithelial cells [80]. This mechanism may explain the presence of phages across body sites even in the absence of disease. However, studies are still few in number, and we lack a full understanding of the consequences that systemic acute or chronic phage exposure may have on human health.

Phages have also been found to play a role in *Clostridium difficile* (*C. diff*) infection. *C. diff* is a bacterial pathogen that causes pseudomembranous colitis with infection often following antibiotic treatment [81]. While it has been understood for decades that commensal bacteria play a role in protecting against colonization, it is not clear what role phages may play in *C. diff* infection and pathogenesis. Phages have been known to induce and alter virulence in bacteria, and *C. diff* is no exception, with several phages being shown to encode virulence factors [82]. Many of the currently known phages infecting *C. diff* have been found in the form of prophages, and some sequenced isolates contain five or more prophages as well as several cryptic plasmids which could be episomal phages [83]. Most of our current knowledge of *C. diff* phages involves studies in which prophages were induced into lytic lifecycles. Combining the knowledge of virulence-encoding phages in *C. diff* and phage induction by antibiotics, one can speculate that it is not merely bacterial depletion playing a role in *C. diff* infection. However, the role of phages in *C. diff* is still poorly understood, and it is not clear if resident phages in the gut could contribute to *C. diff* virulence.

These studies provide broad insight into the potential role phages play in human diseases of the gut. Further, they illustrate the need for continued research into gut phages as well as consideration of phages when designing microbiota studies. Finally, it is important to note that many of these studies were performed on stool samples, which may not fully represent the true diversity of phages in the human gut and do not provide any spatial information about phage distribution along the length of gastrointestinal tract or within intestinal substructures, such as crypts and villi [84]. Stool samples also favor discovery of phages which are either free virions, readily inducible or contained within bacteria that are shed in the stool, which likely does not fully capture all phage. This is an important point, especially when considering FMT, wherein donor stool samples are transferred to recipients as part of treatment for diseases like *C. diff* and IBD. Phages will also be a part of the donor FMT, and thus will play a role in FMT engraftment and success. 

### 4.3. Phages on the Skin

It will come as no surprise that as with many other organ systems, the skin has its own unique repertoire of bacteria and phages. Though a single organ, the skin contains multitudes of spatially organized microbiota. These regions vary both in moisture and exposure to external environments, leading to the development of unique microbial communities. As with the bacteria in these sites, the phages in different skin sites appear to have significant differences in composition [85]. 

Interpersonal variation in the skin phageome is higher than variation over time, however, and the skin phageome is less temporally stable than the phageome at other sites. Though some evidence exists for the variability of the skin virome over time, other studies have found a core phageome in the skin [58]. It also appears that, unlike eukaryotic viruses which were found across many body sites, particular phages had specificity for particular body sites. Correlations between phage and *Propionibacterium acnes* suggestive of an antagonistic relationship have been observed, and analysis of healthy patients and those with acne found increased prevalence and abundance of phages targeting *P. acnes* in healthy patients [59]. Additionally, it appears there is also a correlation with phage abundance and age. Further examination of *P. acnes* phages revealed a strong correlation between specific phages and bacterial strains, highlighting the specificity and diversity of phage–host interactions in the skin [60].

In general, research into the role of phages in human skin disease has not been extensive. One of the few studies examining the lesion microenvironment in patients with psoriasis, a chronic skin inflammatory condition, revealed several phage-derived genes in *Staphylococcus* strains with increased virulence-related genes [61]. Phages targeting *Staphylococcus* have been found to be components of the core phageome in skin [58]. Further exploration will be critical to define interactions between phage and inflammatory skin conditions.

### 4.4. Phages in the Vaginal Microbiota

The vaginal microbiota is unique compared with other body sites, being generally dominated by a single genus of bacteria *Lactobacillus,* at least in healthy Western Caucasians [86]. Unsurprisingly, many *Lactobacillus*-associated phages have also been identified in the vaginal tract, including several prophages [68]. Studies of *Lactobacillus iners,* a strain associated with dysbiosis in the vagina, report an increase in CRISPR expression, indicating a potential resistance to phage infection by some taxa [69]. Even in women with highly distinct vaginal bacterial communities, however, distinct viral community structures were not observed [70]. In patients with bacterial vaginosis (BV), which is usually characterized by the presence of other bacterial taxa, including *Gardnerella*, it has been shown there is an increase in phage load [71]. *Gardnerella* itself carries several prophages [72], but no significant difference in phage induction has been observed between BV patients and healthy controls [73]. In general, however, the role phages play in BV and in the homeostasis of the vaginal microbiota is not well-understood. 

### 4.5. Phages in the Lungs

As in all the other body sites discussed, research is beginning to uncover the role of phages in health and disease of the lungs. Studies in healthy human lungs have shown a resident population of phages in the lower lung, which appears to play a role in the stimulation of immune pathways [56]. A study in samples collected from patients with cystic fibrosis, a genetic disorder characterized by the production of thick mucus that interferes with breathing and organ function, revealed a filamentous bacteriophage that increased the virulence of *Pseudomonas aeruginosa* [57]. It was also shown that patients whose lungs contained this phage were more likely to have chronic *P. aeruginosa* infection, as well as increased resistance to antipseudomonal antibiotics. Analysis of the phage found it allowed bacteria to sequester these antibiotics, revealing a potential role in antibiotic resistance in *P. aeruginosa*.

Across all body sites discussed, it is clear phages are more than merely bystanders in the environment, and they are playing roles in both homeostasis and disease pathogenesis. The presence of “core phageomes” in many body sites argues for conserved roles of phages in the human microbiota. It is also apparent that phage interactions are a key component of dysbiosis, either by regulating or facilitating the invasion of exogenous bacterial taxa. 

## 5. The Future of Phages in Research and Health

The future of phages in research and healthcare is only just beginning to be realized. Outside of the realm of phage therapy, dozens of avenues for phages have opened. From bioengineering to cancer research, applications for phages are numerous. Originally thought of merely as antibiological particles, we now recognize them as bacterial viruses with broad potential. To close, we will briefly discuss areas of human health that have thus far not been explored in regard to phages.

It is often highlighted in research on endogenous phages and even in general microbiota studies that little is known about the human virome. The past decade has revealed a key role for the phages in gut health, but there is limited evidence as to their role at other sites. Many key questions remain regarding phages in both maintenance of a healthy microbiota and their role in disease and dysbiosis. Discussed above is the ability for phages to enter the circulatory system and access nearly any region of the human body. Given this faculty for translocation, we would be remiss to not consider phages when studying disease at nearly any site. 

An obvious potential area of importance for phages is in skin diseases. However, both eczema and psoriasis have limited studies examining the skin virome and even fewer studies examining the virome at sites of skin irritation. Psoriasis, which forms plaques, could yield compelling evidence of phage interactions in disease. Phages have also been neglected in the study of autoimmune diseases. It has now become apparent that infection with eukaryotic viruses increases the risk of several autoimmune diseases [87]. Studies have found that humans develop anti-phage antibodies [88], indicating the potential for phage immunogenicity [89]. However, the effects of phages, both endogenous and those that enter from the environment, have not been examined in these disorders. 

There are many potential therapeutic uses of phages to treat human diseases, and the future of phages is bright. It is clear by the surge in publications and popular press involving phages that we are in the midst of a phage renaissance, and much ongoing work may have dramatic implications for human health and disease. Many areas of human health have been neglected when it comes to phage research, but this is being gradually remedied, and we are entering an unprecedented age of discovery when it comes to phages. Twenty years ago, Roger Hendrix made the statement, “All the world’s a phage” [90], and today, this has never been more true.

## Figures and Tables

**Figure 1 viruses-11-00587-f001:**
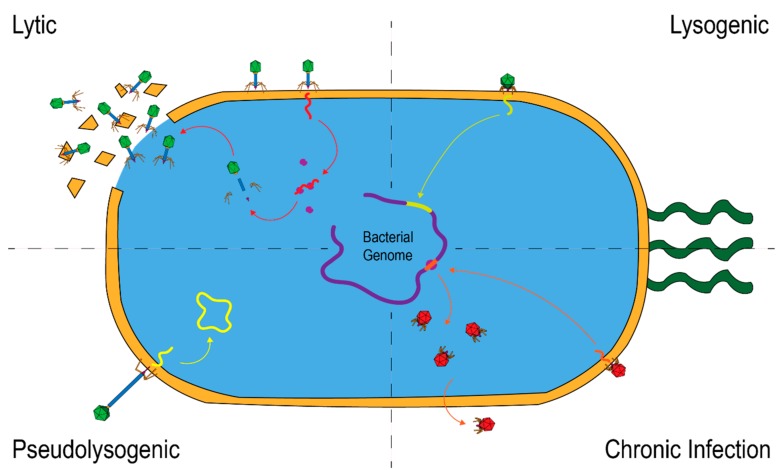
An illustration of the four most well-studied phage lifecycles. Lytic lifecycles involve active replication of the phage and eventual lysis of host bacteria. Lysogenic phages integrate into the host genome or insert as plasmids and do not replicate. Chronic infection produces viral progeny but does not lyse the cell. Phages in a pseudolysogenic state insert DNA, which remains circularized in the cytoplasm.

**Figure 2 viruses-11-00587-f002:**
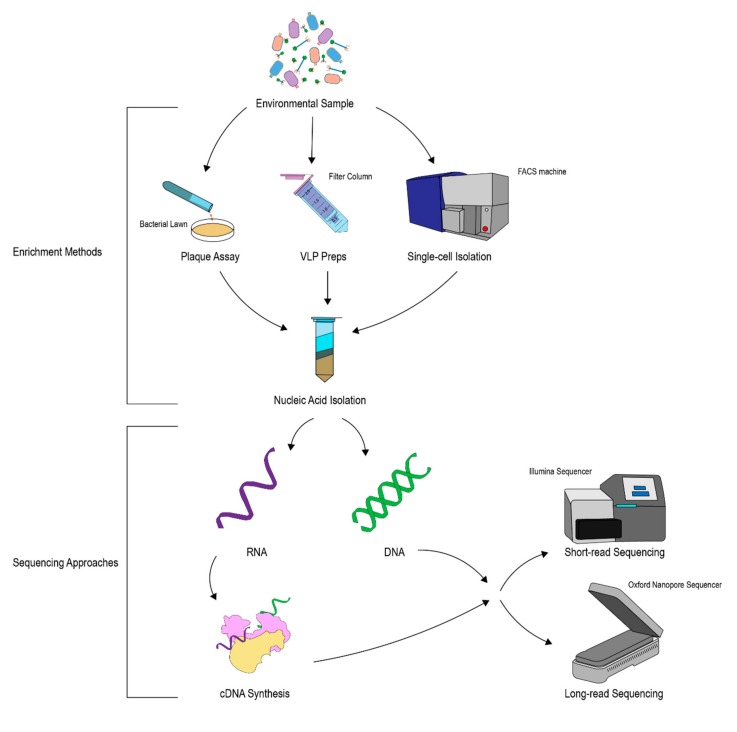
A flowchart describing potential methods for analyzing phages from an environmental sample containing a complex mixture of microbes and viruses. Methods for enriching phages from these samples include plaque assays, virus-like particle preparations, and single-cell isolation. Nucleic acid can then be purified from the enriched samples. The nucleic acid can then be sequenced either by short-read sequencing or long-read sequencing, though generally, a combination of both can produce the best genome assemblies [34].

**Figure 3 viruses-11-00587-f003:**
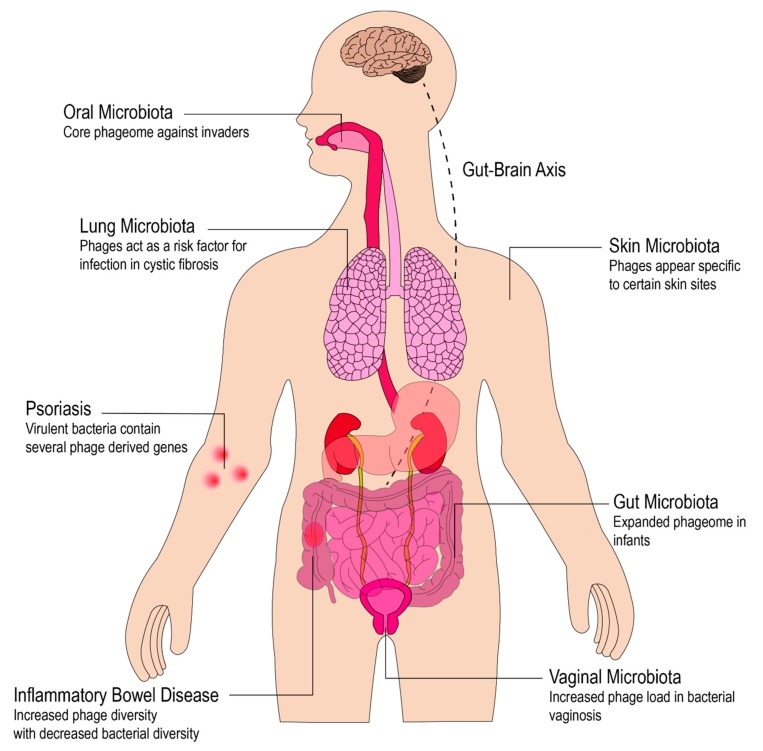
A diagram highlighting regions in the body in which phages have been indicated to play a role. These regions include the oral microbiota [54,55], the lung microbiota [56,57], the skin microbiota [58,59,60,61], the gut microbiota [62,63,64,65,66,67], and the vaginal microbiota [68,69,70,71,72,73].

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
