# Peer review of "Phages and Human Health: More Than Idle Hitchhikers"

_viruses, 2019, doi:10.3390/v11070587_

Reviewer 1 Report

The review submitted deals an important topic: the role of phages in human health. It contains a significant amount of information and a well written section “The Role of Phages in the Human Microbiota”. However, the review as a whole needs much more focusing and improvement.

1.    A supposedly key section (Phage for Therapy) is very short and lacks in-depth discussion of the advantages and limitations of therapeutic phages.

2.    I would like to see more connection between various sections of the review, e.g. phage therapy and human microbiota sections

3.    Some sections (e.g. phage display) are not needed for this particular review.

4.    Phage biology (Section 2) contains a number of errors and gaps in understanding of phages; this should be thoroughly revised. Phage biology is well presented elsewhere (numerous textbook), therefore the whole section may be significantly shorten.

Please see notes below.

Page 2; line 51 – reference is not added.

Page 2; line 54 , remove word “potential” as phages were indeed used at  that time.

Line 71; “phages have been grouped into either lytic and lysogenic life cycles” please revise this sentence.

Lines 75-78; Not correct; temperate phages upon infection can enter either lytic or lysogenic cycles!

Line 85; “Pseudolysogenic phages2 should be pseudolysogenic state.

Line 87; “remain free-standing in a plasmid-like form” that is confusing as some typical temperate phages form lysogens with a prophage in the form of plasmid.

Pseudolysogeny was known for many decades it refers to (possibly) a number of states, many of which are still poorly understood.

Lines 113-114; Isolationof phages using traditional approach is not correctly described. Enrichment techniques should be addressed.

Lines 128-129; “plaque assays only capture phages that are lytic or induced into a lytic lifecycle” Absolutely wrong statement, in fact traditional techniques (plaque assay) is perfectly applicable to lytic, temperate and other phages (including M13 and those with pseudolysogenic tendencies).

Lines 139-140; It is important to note that preparation of viral fractions (prior metagenomic sequencing) requires purification from cellular matter and free NA.

Lines 148-153; These are not limitations of the metagenomic sequencing but rather of existing bioinformatics tools.

Lines 158-159 “This method has been used to capture full bacterial genomes as well as to uncover phages in these cells and isolate phage genomes directly” Not clear how you can uncover phages within cells. Do you mean prophages? If so, any sequencing technology could do the same.

“Single amplified genomes” limitations must be  discussed! These are applicable to the most abundant cells/ viruses only.

Line 175; “non-integrated phage signals” what does that mean; should be explained.

Line 168; “3.2 Computational Analysis of Phages” This title poorly reflect the content of the section which mostly deals with phage identification.

Line 367; It should be noted that d’Herelle founded phage therapy.

Author Response

The review submitted deals an important topic: the role of phages in human health. It contains a significant amount of information and a well written section “The Role of Phages in the Human Microbiota”. However, the review as a whole needs much more focusing and improvement.

We thank the reviewer for their thoughtful comments and insightful points. We have addressed all suggested issues and believe our review is now both more focused and also more complete.

1.    A supposedly key section (Phage for Therapy) is very short and lacks in-depth discussion of the advantages and limitations of therapeutic phages.

We agree with the reviewer that this section did not describe phage therapy with sufficient detail and have chosen to remove it as it is generally outside the scope of our review.

2.    I would like to see more connection between various sections of the review, e.g. phage therapy and human microbiota sections

Related to comment #1, we agree with the reviewer and have removed the phage therapy section. The sections of our review are now more clearly interconnected.

3.    Some sections (e.g. phage display) are not needed for this particular review.

We agree with the reviewer and have removed phage display from the review.

4.    Phage biology (Section 2) contains a number of errors and gaps in understanding of phages; this should be thoroughly revised. Phage biology is well presented elsewhere (numerous textbook), therefore the whole section may be significantly shorten.

We agree with the reviewer. Phage biology is a complex and well-developed field spanning decades of research. However, we disagree that this section requires significant revision as it is presented only as a brief overview. We have added a sentence to reaffirm that this is not meant to be comprehensive review of phage biology and readers can seek additional information elsewhere.

“Here we present a brief overview of phage biology; a number of reviews and other sources have covered this topic in greater depth”

Page 2; line 51 – reference is not added.

We have now added the appropriate reference.

S. Minot, A. Bryson, C. Chehoud, G. D. Wu, J. D. Lewis, and F. D. Bushman, “Rapid evolution of the human gut virome,” Proc. Natl. Acad. Sci., vol. 110, no. 30, pp. 12450–12455, 2013.”

Page 2; line 54 , remove word “potential” as phages were indeed used at  that time.

We have removed the use of “potential” here.

Line 71; “phages have been grouped into either lytic and lysogenic life cycles” please revise this sentence.

We have now added the word “traditionally”, to indicate that early phage research classified most phages as lytic or lysogenic. We then explain in the following section that additional phage life cycles have since been identified.

“Traditionally, phages have been grouped into either lytic or lysogenic life cycles

Lines 75-78; Not correct; temperate phages upon infection can enter either lytic or lysogenic cycles!

This sentence has been rephrased to make it more clear, and the term “temperate” has been removed.

“During infection, lysogenic phages instead incorporate their genome into that of their bacterial host, or may maintain it as a plasmid [16], where it remains and is passively replicated when the bacterium replicates. These phages are referred to as “prophages.”

Line 85; “Pseudolysogenic phages2 should be pseudolysogenic state.

We have made the suggested change for clarity.

Line 87; “remain free-standing in a plasmid-like form” that is confusing as some typical temperate phages form lysogens with a prophage in the form of plasmid.

Pseudolysogeny was known for many decades it refers to (possibly) a number of states, many of which are still poorly understood.

We have addressed this by removing the comparison of pseudolysogenic phage to plasmids and adding a concrete statement clarifying this point.

“Pseudolysogenic phages have been described as being in a state of “stalled development”, and can either be incorporated into the host genome or remain as free DNA in the cytoplasm [17]. This free DNA is distinct from the plasmid form of some lysogenic phages [20].”

Lines 113-114; Isolationof phages using traditional approach is not correctly described. Enrichment techniques should be addressed.

 We agree this was lacking in the description and have added a sentence to indicate enrichment is often needed as well as citing an enrichment protocol.

“ Often phages in suspension must be enriched, which can be done by inoculating a liquid culture of bacteria with the suspected phage and allowing the culture to clarify [25]. “

Lines 128-129; “plaque assays only capture phages that are lytic or induced into a lytic lifecycle” Absolutely wrong statement, in fact traditional techniques (plaque assay) is perfectly applicable to lytic, temperate and other phages (including M13 and those with pseudolysogenic tendencies).

We agree that our presentation of plaque assays here was not adequately clear. We have rewritten the section to indicate that phages of any lifecycle can be captured with this assay, given proper conditions.

“Additionally, plaque assays are only able to capture phages which form plaques, which may require additional treatments, such as mitomycin C, to induce phage into a lytic state [29].”

Lines 139-140; It is important to note that preparation of viral fractions (prior metagenomic sequencing) requires purification from cellular matter and free NA.

 We have added a sentence clarifying that VLP methods remove bacteria and free NA.

 “ VLP preparation is a broad descriptor which generally involves removal of bacteria and free nucleic acid, thereby enriching for viral particles, which are then broken open to isolate DNA or RNA.”

Lines 148-153; These are not limitations of the metagenomic sequencing but rather of existing bioinformatics tools.

 We agree with the reviewer that our statements are more reflective of the limitations of bioinformatics tools and have revised this section to make that more clear.

“ … though these issues may be addressed by improvements to databases and search and assembly algorithms”

Lines 158-159 “This method has been used to capture full bacterial genomes as well as to uncover phages in these cells and isolate phage genomes directly” Not clear how you can uncover phages within cells. Do you mean prophages? If so, any sequencing technology could do the same.

 In this case, we are indicating recovery of non-integrated phage either in the cytoplasm of the bacterial cell or adhered to the cell surface. This was unclear and has been revised to make specific statements on the ability to extract DNA from intact virion associated with the bacteria.

“ This method has been used to capture full bacterial genomes as well as to uncover phage virions in the cytoplasm or adhered to the surface of these cells and thereby isolate phage genomes directly [35], [37].”

“Single amplified genomes” limitations must be  discussed! These are applicable to the most abundant cells/ viruses only.

 We agree with the reviewer that many applications of SAGs are only able to find the more abundant cells/viruses, but note that the cited papers applying SAGs have been able to find microbes present at low abundance. However, we have added a sentence acknowledging that SAGs are biased towards recovery of organisms at higher abundance.

 “ As with any method analyzing complex communities, current SAG techniques are biased toward recovery of more common community members.”

Line 175; “non-integrated phage signals” what does that mean; should be explained.

 This has been changed to “non-integrated phage sequences” for precision and clarity.

Line 168; “3.2 Computational Analysis of Phages” This title poorly reflect the content of the section which mostly deals with phage identification.

We have retitled this section to: “Computational Identification of Phages”.

Line 367; It should be noted that d’Herelle founded phage therapy.

The section on phage therapy has now been removed.

Reviewer 2 Report

The manuscript by Lawrence at all is an interesting short review summarizing current methods of phage discovery and characterization, and the results of recent studies concerning the role of phages in shaping human microbiome, and hence influencing the development of certain diseases. The manuscript is mostly up to data and of general interest. Thus, it should be published. I have only some suggestions concerning correction that are necessary to introduce to avoid confusing the readers and to improve the manuscript text. They are listed below.

L.76: Prophages of certain temperate phages (e.g. P1 and N15) exist in cells in the form of plasmids. This should be mentioned. The same applies to Figure 1.

L. 87. The description of pseudolysogeny provided according to reference 13 is a huge oversimplification. Additionally it suggests misleadingly that pseudolysogeny and lysogeny in a form of plasmid is the same. This should be corrected to avoid confusing the readers.  (see e.g.  Hobbs and Abedon, 2016. FEMS Microbiol. Lett. Apr;363(7). pii: fnw047. doi: 10.1093/femsle/fnw047 and references therein; Cenens et al., 2013. Bacteriophage Jan 1; 3(1): e25029, doi: 10.4161/bact.25029 ). What about phage carrier state? It should be mentioned by the authors too.  

L. 95. Replace "genetic sequence identity" with "genetic and protein sequence homologies"

L. 96-97. Current taxonomy within Caudovirales is slightly more complex and includes two additional families: Herelleviridae and Ackermannviridae. The information concerning families within the Caudovirales order should be updated by the authors (see e.g. https://talk.ictvonline.org/files/master-species-lists/m/msl/8266 to download the recent version of ICTV master species list or refer to publications)

L. 115. Replace "holes" with "clearings"

L. 121. Remove first "as well"

Figure 2. Oxford nanopore sequencer generates sequence data with only about 90% accuracy. Thus it can provide scaffolds for the assembly of short reads, but cannot be used separately to determine complete genomic sequences. This should be taken into consideration by the authors. Perhaps "+" in Figure 2 between Illumina sequencer and Oxford Nanopore Sequencer will suffice.

L. 173. Add "PHASTER" to the list of programs for prophage searches.

L. 292. Replace "are prophages" with " have been found in the form of prophages"

L. 333-334. The eczema treatment mentioned by the authors is not based on phages but on phage lytic protein, and it does not indicate a potential role for phages in eczema pathogenesis. It simply employs bacteriolytic activity of phage enzyme, targeting specifically Staphylococcus aureus to decrease the number of S. aureus cells that may cause eczema. This should be corrected by the authors.

L. 378-387. There are several reports indicating the safety of phage application in humans. They should be cited here as well.

L. 381-382. Changes in phage specificity, especially redirecting phages to bacteria of different taxons, are not as frequent as the acquisition of phage resistance by bacteria. Thus, if the authors consider changes in phage specificity as a risk they should provide a reference to prove it.

L. 394-408. The chapter concerning phage display does not fit to the rest of the manuscript text and should be removed. Phage display is a specific technique that serves to discover or identify peptides specifically interacting with certain organic or inorganic compounds, including even tissue specific receptors inside a body (cancer cell-specific receptors among them). However, it does not serve to treat diseases but e.g., to discover peptides (as such or as parts of proteins, including antibodies) that may be used for such treatment. Thus, this chapter has no relation with the main body of this manuscript. It could be included in a review concerning biotechnological applications of phages.  

Author Response

The manuscript by Lawrence at all is an interesting short review summarizing current methods of phage discovery and characterization, and the results of recent studies concerning the role of phages in shaping human microbiome, and hence influencing the development of certain diseases. The manuscript is mostly up to data and of general interest. Thus, it should be published. I have only some suggestions concerning correction that are necessary to introduce to avoid confusing the readers and to improve the manuscript text. They are listed below.

We thank the reviewer for their positive response to our manuscript and have addressed all of their recommendations. The manuscript is now much improved.

L.76: Prophages of certain temperate phages (e.g. P1 and N15) exist in cells in the form of plasmids. This should be mentioned. The same applies to Figure 1.

We agree with the reviewer that we neglected to mention prophages in plasmids and have added a sentence and reference to the section mentioning the ability of temperate phages to enter their host as a plasmid.

 “ During infection, lysogenic phages instead incorporate their genome into that of their bacterial host, or may maintain it as a plasmid [16] where it remains and is passively replicated when the bacterium replicates.”

L. 87. The description of pseudolysogeny provided according to reference 13 is a huge oversimplification. Additionally it suggests misleadingly that pseudolysogeny and lysogeny in a form of plasmid is the same. This should be corrected to avoid confusing the readers.  (see e.g.  Hobbs and Abedon, 2016. FEMS Microbiol. Lett. Apr;363(7). pii: fnw047. doi: 10.1093/femsle/fnw047 and references therein; Cenens et al., 2013. Bacteriophage Jan 1; 3(1): e25029, doi: 10.4161/bact.25029 ). What about phage carrier state? It should be mentioned by the authors too. 

We agree we did not adequately describe pseudolysogeny as well as we could have and have reformatted the section to make the lifecycle description more clear and have cited the above references.

“Pseudolysogenic phages have been described as being in a state of “stalled development”, and can either be incorporated into the host genome or remain as free DNA in the cytoplasm [17]. This free DNA is distinct from the plasmid form of some lysogenic phages [20]. Their genetic material is inserted into the host cell, but rather than incorporating into the host genome, the genetic material remains free in the cytosol. These phages can undergo lytic or lysogenic lifestyles depending on environmental conditions. Pseudolysogenic phages may also be referred to as being in a phage “carrier state” [21][22]

In regard to carrier state we mention it in the above passage and have chosen to present carrier state according to the definition as defined by Stephen Abedon in 2009; “pseudolysogeny can result from phage maintenance within a culture via lytic infection of only a portion of the bacteria present, a phenomenon which I refer to as a phage carrier state”

L. 95. Replace "genetic sequence identity" with "genetic and protein sequence homologies"

We agree genetic sequence identity was a poor statement and have made this revision.

L. 96-97. Current taxonomy within Caudovirales is slightly more complex and includes two additional families: Herelleviridae and Ackermannviridae. The information concerning families within the Caudovirales order should be updated by the authors (see e.g. https://talk.ictvonline.org/files/master-species-lists/m/msl/8266 to download the recent version of ICTV master species list or refer to publications)

We agree with the reviewer that mentioning the more recent additions to Caudovirales  is important and we have revised the section appropriately.

“The most well-classified order of phages, because these were among the first to be carefully studies, is Caudovirales which is further divided into five families, the three most well-characterized being Myoviridae, Siphoviridae, and Podoviridae. There are also two more recently described families; Ackermannviridae and Herelleviridae [24], [25].”

L. 115. Replace "holes" with "clearings"

We agree this is better phrasing and have made this change.

L. 121. Remove first "as well"

This was removed.

Figure 2. Oxford nanopore sequencer generates sequence data with only about 90% accuracy. Thus it can provide scaffolds for the assembly of short reads, but cannot be used separately to determine complete genomic sequences. This should be taken into consideration by the authors. Perhaps "+" in Figure 2 between Illumina sequencer and Oxford Nanopore Sequencer will suffice.

While we agree with the reviewer that Nanopore data can have high error rates, recent publications have illustrated that Nanopore sequencing alone can assemble full genomes with very high accuracy. (Loman et al. 2015).

Loman NJ, Quick J, Simpson JT. A complete bacterial genome assembled de novo using only nanopore sequencing data. Nat Methods. 2015;12(8):733–5.

However, we agree that for many applications addition of short-read data to long-read data would benefit genomic assembly. This was highlighted by a recent paper describing meta-virome analysis (https://www.ncbi.nlm.nih.gov/pubmed/31086738). We have added a sentence and reference to inform readers of the challenges.

“ The nucleic acid can then be sequenced either by short-read sequencing or long-read sequencing, though generally a combination of both can produce the best genome assemblies [33]"

L. 173. Add "PHASTER" to the list of programs for prophage searches.

We have added this program to the list.

L. 292. Replace "are prophages" with " have been found in the form of prophages"

We agree this is a better way to state this and have made this change.

L. 333-334. The eczema treatment mentioned by the authors is not based on phages but on phage lytic protein, and it does not indicate a potential role for phages in eczema pathogenesis. It simply employs bacteriolytic activity of phage enzyme, targeting specifically Staphylococcus aureus to decrease the number of S. aureus cells that may cause eczema. This should be corrected by the authors.

We agree with the reviewer that the phrasing for these treatments was poor. We have removed this sentence as we have generally removed phage therapy sections based upon comments by reviewer #1.

L. 378-387. There are several reports indicating the safety of phage application in humans. They should be cited here as well.

Feedback from other reviewers also found this section lacking, and we have chosen to remove the phage therapy section entirely.

L. 381-382. Changes in phage specificity, especially redirecting phages to bacteria of different taxons, are not as frequent as the acquisition of phage resistance by bacteria. Thus, if the authors consider changes in phage specificity as a risk they should provide a reference to prove it.

As above, we have chosen to remove this section.

L. 394-408. The chapter concerning phage display does not fit to the rest of the manuscript text and should be removed. Phage display is a specific technique that serves to discover or identify peptides specifically interacting with certain organic or inorganic compounds, including even tissue specific receptors inside a body (cancer cell-specific receptors among them). However, it does not serve to treat diseases but e.g., to discover peptides (as such or as parts of proteins, including antibodies) that may be used for such treatment. Thus, this chapter has no relation with the main body of this manuscript. It could be included in a review concerning biotechnological applications of phages. 

We agree this section was out of place in this review and have removed it.

Reviewer 3 Report

Lawrence et al. summarize in this review the current state of knowledge about bacterial viruses, their biology, metagenomics analysis, potential role as components of human microbiome and the possibilities of their application in scientific research and human health.

The paper is well structured and written; clear, precise, and easy to understand.  It also addresses a subject that is of great interest in the scientific community. The topic is in line with the "Viruses" journal and therefore I recommend publishing after minor revisions.

Line 77: the term "temperate" requires specification for the prophage type (integrated or maintained extrachromosomally with either circular or linear genomes)

Lines 85-87 term "pseudolysogeny" needs to be clarified. Indeed, pseudolysogeny can be defined as the stage of stalled development of a bacteriophage in a host cell without either multiplication of the phage genome (as in lytic development) or its replication synchronized with the cell cycle and stable maintenance in the cell line (as in lysogenization), which proceeds with no viral genome degradation, thus allowing the subsequent restart of virus development. It should be stressed that this phenomenon is usually caused by unfavorable growth conditions for the host cell (such as starvation) and is terminated with initiation of either true lysogenization or lytic growth when growth conditions improve.

Line 54: there is no reference

Author Response

Lawrence et al. summarize in this review the current state of knowledge about bacterial viruses, their biology, metagenomics analysis, potential role as components of human microbiome and the possibilities of their application in scientific research and human health.

The paper is well structured and written; clear, precise, and easy to understand.  It also addresses a subject that is of great interest in the scientific community. The topic is in line with the "Viruses" journal and therefore I recommend publishing after minor revisions.

We thank the reviewer for their positive comments and have addressed all suggested alterations.

Line 77: the term "temperate" requires specification for the prophage type (integrated or maintained extrachromosomally with either circular or linear genomes)

Based on this feedback and similar concerns by the other reviewers, we have rewritten this section.

“When sufficient numbers of progeny are produced, the host cells lyse, generally killing the host in the process. In contrast, lysogenic phages undergo very little replication in their host. During infection, lysogenic phages instead incorporate their genome into that of their bacterial host, or may maintain it as a plasmid [16] where it remains and is passively replicated when the bacterium replicates. These phages are referred to as “prophages.”"

Lines 85-87 term "pseudolysogeny" needs to be clarified. Indeed, pseudolysogeny can be defined as the stage of stalled development of a bacteriophage in a host cell without either multiplication of the phage genome (as in lytic development) or its replication synchronized with the cell cycle and stable maintenance in the cell line (as in lysogenization), which proceeds with no viral genome degradation, thus allowing the subsequent restart of virus development. It should be stressed that this phenomenon is usually caused by unfavorable growth conditions for the host cell (such as starvation) and is terminated with initiation of either true lysogenization or lytic growth when growth conditions improve.

 We have addressed the reviewer’s comment by including an introductory line making it clear this is a broad overview and that additional details are available via other references. We have also improved the pseudolysogeny section by briefly alluding to the suggested details.

“Here we present a brief overview of phage biology; a number of reviews and other sources have covered this topic in greater depth [14], [15]

“Pseudolysogenic phages have been described as being in a state of “stalled development”, and can either be incorporated into the host genome or remain as free DNA in the cytoplasm [17]. This free DNA is distinct from the plasmid form of some lysogenic phages [20]. Their genetic material is inserted into the host cell, but rather than incorporating into the host genome, the genetic material remains free in the cytosol. These phages can undergo lytic or lysogenic lifestyles depending on environmental conditions. Pseudolysogenic phages may also be referred to as being in a phage “carrier state” [21][22]"

Line 54: there is no reference

We have now added in the reference.

S. Minot, A. Bryson, C. Chehoud, G. D. Wu, J. D. Lewis, and F. D. Bushman, “Rapid evolution of the human gut virome,” Proc. Natl. Acad. Sci., vol. 110, no. 30, pp. 12450–12455, 2013.”